# Microstructure Evolution Mechanism of W_f_/Cu_82_Al_10_Fe_4_Ni_4_ Composites under Dynamic Compression at Different Temperatures and Strain Rates

**DOI:** 10.3390/ma14195563

**Published:** 2021-09-25

**Authors:** Zhe Wu, Yang Zhang, Haifeng Jiang, Shuai Zhao, Qingnan Wang

**Affiliations:** 1College of Mechanical and Electrical Engineering, Northeast Forestry University, Harbin 150040, China; jhf15776821124@163.com (H.J.); a782883866@163.com (S.Z.); wqn0203@163.com (Q.W.); 2College of Science, Northeast Forestry University, Harbin 150040, China

**Keywords:** W_f_/Cu_82_Al_10_Fe_4_Ni_4_ composites, microstructure, high temperature, dynamic compression

## Abstract

W_f_/Cu_82_Al_10_Fe_4_Ni_4_ composites were fabricated by the pressure infiltration method. The composites were compressed by means of a split Hopkinson pressure bar (SHPB) with strain rates of 800 and 1600 s^−1^ at different temperatures. The microstructure of the composites after dynamic compressing was analyzed by transmission electron microscopy (TEM). Observation revealed that there were high-density dislocations, stacking faults, twins, and recrystallization existing in the copper alloy matrix of the composites. High-density dislocations, stacking faults, and twins were generated due to the significant plastic deformation of the copper alloy matrix under dynamic load impact. We also found that the precipitated phase of the matrix played a role in the second phase strengthening; recrystallized microstructures of copper alloy were generated due to dynamic recrystallization of the copper alloy matrix under dynamic compression at high temperatures.

## 1. Introduction

With the development of defense technology, the requirements for various properties of armor piercing projectiles are also increasing, especially the mechanical properties of materials used in armor-piercing projectiles [1,2,3,4,5]. Metal matrix composites are generally formed by the reasonable combination of one material as the matrix and the other material as the reinforcing phase, so they often have better properties than the single materials. Therefore, in recent years, many countries began to study composite armor-piercing projectiles to improve their penetration ability [6,7,8]. Because tungsten has the advantages of high density and high melting point (3410 °C), it is often used to prepare armor-piercing projectiles [9,10,11,12,13,14,15]. Bulk tungsten is usually made by sintering fine tungsten particles together, while tungsten fiber is obtained by stretching the bulk tungsten at high temperature. In the drawing process, the small particles inside the tungsten fiber are stretched into fibrous shapes, so the tungsten fiber can obtain higher mechanical properties [16,17,18,19,20]. In view of the above advantages, tungsten fiber was selected as the reinforcement of a new generation of composites for armor-piercing projectiles [19]. In the selection of the matrix alloy, copper alloy has the characteristics of high density and high strength, tungsten copper composites have been successfully applied in many industrial fields, and the preparation technology is relatively mature. Therefore, copper alloy was selected as the matrix of composites in this study [21,22,23,24,25]. To improve the interfacial bonding strength, iron, nickel, and aluminum were added to copper to increase the interfacial strength. Then, W_f_/Cu_82_Al_10_Fe_4_Ni_4_ composites were prepared by a pressure infiltration method [26,27,28,29,30].

Regarding the process of flying and impacting of the armor piece, as the body temperature increases sharply, research on the microstructures of the armor-piercing projectile under high-temperature conditions is necessary. By observing the microstructure morphology of the material after dynamic compression, the microstructure change and failure behavior of the material under high strain rate can be analyzed. At the same time, the microevolution mechanism of materials can be explained, thus laying a foundation for the subsequent improvement of material properties. While research on this aspect is scarce at present, the dynamic compression of W_f_/Cu_82_Al_10_Fe_4_Ni_4_ composites under the condition of different temperatures and strain rates was achieved using a split Hopkinson Pressure bar (SHPB). The SHPB was prepared by the China University of Science and Technology (He Fei, China). The microstructure of the composites was investigated by transmission electron microscopy (TEM), and the TEM model was an FEI G2F30 (FEI, Hillsborough, OR, USA).

## 2. Materials and Experimental Procedure

W_f_/Cu_82_Al_10_Fe_4_Ni_4_ composites were prepared by pressure infiltration method [31,32,33]. The preparation of W_f_/Cu_82_Al_10_Fe_4_Ni_4_ composites included the surface treatment of tungsten fiber, the melting of the matrix copper alloy, the preparation of a mold, the preparation of composites, and so on. Cu_82_Al_10_Fe_4_Ni_4_ alloy (Harbin Far East Copper Co., Ltd, Harbin, China) was selected as the matrix of the composite. The matrix was melted by electromagnetic stirring under vacuum conditions to ensure a uniform structure and accurate composition. Straightened tungsten fiber (Xiamen Honglu Tungsten Molybdenum Industry Co., Ltd, Xia Men, China) with a diameter of 0.25 mm was selected as the reinforcement. The tungsten fibers were cut in 100 mm, soaked in 40% HF liquor solution (Harbin Boda Chemical Reagent Distribution Co., Ltd, Harbin, China), and the surface oxide film was removed. The tungsten fibers with pure surfaces were obtained by ultrasonic cleaning in acetone (Harbin Boda Chemical Reagent Distribution Co., Ltd, Harbin, China) and alcohol (Harbin Boda Chemical Reagent Distribution Co., Ltd, Harbin, China), sequentially, and the prepared tungsten fibers were vertically inserted into a clean quartz tube. Then, the mold needed to be prepared. The mold included an inner mold, sleeve, and outer mold. The role of the inner mold is to bear the pressure of the press head; the sleeve facilitates demolding; the outer mold protects the sleeve during pressurization. They were all processed with high-strength graphite.

The preparation process of the composite was as follows: the quartz tube was filled with tungsten fibers into the hole of the inner mold, the melted cylindrical master alloy was placed on the upper part of the inner mold, and the preparation was then completed in the vacuum pressure infiltration furnace (Harbin Institute of Technology, Harbin, China). After demolding, W_f_/Cu_82_Al_10_Fe_4_Ni_4_ composites with 80% volume fraction of tungsten fiber were obtained. The preparation was as follows: the entire mold was heated to 1250 °C under vacuum for 30 min, then slowly pressurized to 10 MPa. The molten Cu_82_Al_10_Fe_4_Ni_4_ alloy was pushed into the gap between tungsten fibers, then cooled.

SHPB was selected as the test equipment, and a controllable heating device (China University of Science and Technology, He Fei, China) was installed on the SHPB to control the test temperature. The length of the input and output rods of SHPB was 1000 mm. The length of the impact rod was 150 mm, and the diameter of the impact rod was 12.7 mm. The compression specimen was a cylinder with a diameter of 4 mm and a length of 4 mm. The signals of the incident wave, reflected wave, and transmitted wave were collected through a strain gauge attached between the input rod and the output rod. The signals were input into the dynamic strain instrument (China University of Science and Technology, He Fei, China) for amplification, then input into the oscilloscope (Changzhou Tonghui Electronics Co., Ltd, Chang Zhou, China) to store the signals. The strain gauge measures the waveform change of the input rod and output rod, converts the current signal into a digital signal, and finally obtains the stress–strain data value. According to the data value, the stress–strain curve can be drawn [34]. In this study, test strain rates were 800 and 1600 s^−1^, and the temperatures were 20, 200, 400, and 600 °C. Each group test was tested three times to ensure reliability of the test data. The microstructures of W_f_/Cu_82_Al_10_Fe_4_Ni_4_ composites after dynamic compression were then observed by TEM.

## 3. Results and Discussion

### 3.1. Microstructure of Matrix with a Strain Rate of 800 s^−1^

Figure 1a,b shows that high-density dislocations appeared after dynamic compression with a strain rate of 800 s^−1^. The density dislocations can be attributed to the following reasons: Firstly, the Cu matrix experiences giant plastic deformation because of the extrusion of tungsten fibers during the dynamic compression. As dislocation slipping is the main plastic deformation mechanism of the copper alloys, the dislocation density increases after dynamic compression. Secondly, we found a large amount of precipitates in the composites such (Fe, Ni), AlFe, Al_3_Ni, and Cu_3_Al in previous studies [29,30]. The (Fe, Ni), AlFe, and Al_3_Ni would hinder the dislocation slip due to the pile-up effect, thus leading to the increase in the dislocation density [35]. A high content of W_f_-Cu interface can further hinder the dislocation movement and increase the density of dislocation. Furthermore, as shown in Figure 1c,d, the dislocation density of the matrix alloys after dynamic compression with a strain rate of 1600 s^−1^ increased significantly compared with that with a strain rate of 800 s^−1^.

### 3.2. Microstructure of Matrix with a Strain Rate of 1600 s^−1^

#### 3.2.1. Stacking Fault

Both dislocations and stacking faults occurred in the matrix when the strain rate raised to 1600 s^−1^; Figure 2a shows that the stacking fault spacing was less than 100 nm. There were some experimental errors in the test, and the error range was within 20 nm. Moreover, high-density dislocations tangled around the stacking faults and even some stacking faults were cut off by dislocation arrays, as indicated by the arrows in the figure.

Figure 2b depicts the high-resolution electron microscope (HREM) (FEI, Hillsborough, OR, US) morphology, and Figure 2b shows the atomic structure of stacking faults as marked in Figure 2a. Clear lattice distortion can be observed, and the thickness of stacking fault was about 2 nm. The microstructures of the W_f_/Cu_82_Al_10_Fe_4_Ni_4_ composites after dynamic compression with strain rates of 1600 s^−1^ at room temperature indicated that dislocations were still the main form of matrix copper alloy deformation. Although there were some stacking faults with strain rates of 1600 s^−1^, the stacking fault density was not high and the composites’ deformation was dominated by dislocation slip.

Figure 3a shows the microstructure of W_f_/Cu_82_Al_10_Fe_4_Ni_4_ composites after dynamic compression at 200 °C with a strain rate of 1600 s^−1^. As can be observed, the defects of the composites were dominated by high-density dislocations and stacking faults, the stacking fault spacing was less than 100 nm, and high-density dislocations existed between stacking faults. Figure 3b depicts the selected area diffraction pattern (SADP) from Figure 3a. By calibration of the diffraction in the region, as shown in Figure 3b, the structure in the region was copper after calibration and the crystal zone axis was [110]. Figure 3c shows the microstructure of W_f_/Cu_82_Al_10_Fe_4_Ni_4_ composites after dynamic compression at 400 °C with a strain rate of 1600 s^−1^. As can be observed, damages of composites were dominated by high-density dislocations and stacking faults. The stacking fault thickness was 10 nm, there were some experimental errors in the test, and the error range was about 5 nm. The stacking fault length was 300–500 nm, and the error range was about 100 nm. The stacking fault spacing was 50–100 nm, and the error range was about 20 nm. By calibration of the diffraction spots in the region, as shown in Figure 3d, copper diffraction patterns existed in the region and the crystal zone axis was [211]. Figure 3e shows the microstructure observation of W_f_/Cu_82_Al_10_Fe_4_Ni_4_ composites after dynamic compression at 600 °C. As can be observed, the damage to the composites was dominated by high-density dislocations and stacking faults. The stacking fault thickness was 20 nm, and the stacking fault length was 100–300 nm. The stacking fault spacing was less than 50 nm, and high-density dislocations existed between stacking faults. By calibration of the diffraction spots in the region, as shown in Figure 3f, Cu diffraction existed in the region and the crystal zone axis was [110]. By analysis of Figure 3e,f, it could be learned that dislocations were mainly in two crystal planes of ($\bar{1}11$) and ($\bar{1}1\bar{1}$).

By means of microstructure observation of the matrix alloys after dynamic compression of W_f_/Cu_82_Al_10_Fe_4_Ni_4_ composites at high temperature, we found that with high temperatures and high strain rates, high-density dislocations and stacking faults existed in the matrix copper alloys, indicating that dislocations and stacking faults were the main mechanisms for dynamic compression deformation of copper alloys at high temperatures.

We also found that the dislocation density, as well as the thickness of stacking fault, increased with the rising in the test temperature. The observed phenomenon in the test was that the density of stacking faults increased with the test temperature increasing, related to the plastic deformation of the composites. With the test temperature increasing, the composite plastic deformation would greatly improve, causing the rise in the stacking fault density in the matrix copper alloys. The occurrence of large numbers of stacking faults was related to the large amounts of precipitates in the matrix. As indicated by the arrows in Figure 3a,c,e, there were precipitates in the regions where stacking faults existed, and the stacking fault density was very high in the regions where large amounts of precipitates existed (as shown in Figure 4a,b). The occurrence of large numbers of stacking faults was also related to the large numbers of interfaces as a result of the high-volume fraction of the strengthening phases in the composites. Because precipitates in the matrix and the reinforced tungsten fibers were large-dimensional and considered to be non-deformable particles, and dislocations formed as a result of high strain rate deformation could only interact with the second phase by means of circumvention, high-density dislocations piled around the precipitates, and strengthening phases and accumulation of energy occurred. Thus, when the energy exceeded the stacking fault energy of copper alloys, many stacking faults would occur. According to Zhang’s research [36], stacking fault energy is also an important factor affecting the change in the stacking fault density. The stacking fault energy of Cu, Al, and Ni decreases with the increase in temperature, leading to the increase in the stacking fault density at high temperatures.

Studies on the stacking faults of W_f_/Cu_82_Al_10_Fe_4_Ni_4_ composites formed after dynamic compression at 600 °C were carried out to analyze the stacking fault formation mechanism of the copper matrix under dynamic compression at high temperatures. After compressive deformation with a high strain rate of 1600 s^−1^ and at a high temperature of 600 °C, there were high-density dislocations piling around the precipitates and interfaces of the matrix. When the energy exceeded the stacking fault energy of copper alloys, stacking faults occur [37,38]. The occurrence processes are shown in Figure 3e. When the direction of the external force was the [$\bar{1}12$] direction of the grain, the atoms at the ($\bar{1}1\bar{1}$) layer of the grain would slide [$\bar{1}12$]/6 along the [$\bar{1}12$] direction, the atoms at each layer would move one position layer-by-layer, and the stacking order would change to form a stacking fault. Since the atoms at any layer ($\bar{1}1\bar{1}$) of stacking fault could slip under the external force of the [$\bar{1}12$] direction, at the moment of external force, several stacking faults would be formed at the same time. With the progress of deformation, the grain orientation changed. When the direction of the external force was the [$1\bar{1}2$] direction, the atoms at the layer ($\bar{1}11$) would slip [$1\bar{1}2$]/6 along the [$1\bar{1}2$] direction to form the stacking fault. By these means, the cross-morphology of stacking faults would be formed in two directions in a grain.

#### 3.2.2. Twins

New defeats twins appeared when the dynamic compression temperature rose to 400 °C. As shown in Figure 5a, after dynamic compression at 400 °C, there were deformed twins existing in some regions of the matrix, and the width of the twins was 50–100 nm. Additionally, there were high-density dislocations piling in the twins, and Figure 5b depicts the diffraction spots of Figure 5a. Meyers [35] first proposed that twins would form in copper under impact loading. Deformed twins were formed by means of twins, and twinning is a nucleation and growth process, which generally requires greater stresses. The twin was subject to sudden outbreak at breakneck speed and then twin slices were formed and extended. After compression of W_f_/Cu_82_Al_10_Fe_4_Ni_4_ composites at high temperatures, large plastic deformation occurred in the composites and caused the sharp increase in the dislocation density in the matrix. Meanwhile, the obstacles of precipitates and interfaces in the matrix to the dislocation slip led to the further pile-up of dislocations in local regions, resulting in the emergence of a high-energy region and the occurrence of twins. The formation of twins could significantly improve the plastic deformability of the composites, as twins could change the grain orientation. This was not conducive to slip by means of twinning to induce a new slip system start and the occurrence of new twins.

As indicated by arrows in Figure 6a,b, after dynamic compression at 600 °C, it could also be observed that there were annealing twin structures existing inside and around the grains. The annealing twins appeared and developed together with the recrystallized grains and were mainly inside and around the recrystallized grains.

#### 3.2.3. Dynamic Recrystallization

As shown in Figure 7a, after dynamic compression at 600 °C and a strain rate of 1600 s^−1^, there were many large grains existing in local areas of the W_f_/Cu_82_Al_10_Fe_4_Ni_4_ composite sample, and the grain sizes were 200–500 nm. By calibration of the diffraction spots in this region, as shown in Figure 7b, the diffraction spots in the region were formed by continuous diffraction rings. After calibration, we found that the recrystallized grains of copper appeared in the region, indicating that dynamic recrystallization occurred in the copper at a high temperature and high strain rate. As indicated by the arrows in Figure 7a, it can also be observed that there were high-density dislocations existing inside the recrystallized grains. There were two temperature condition sources for dynamic recrystallization of the composites. One was the high test temperature, and the other was that the adiabatic temperature rise process under high strain rate impact conditions resulted in the temperature rise inside the composites [39]. As shown in Figure 8a, from which the peripheral region of recrystallization could be observed, a large number of cellular dislocations occurred. The dislocation cell sizes were 200–500 nm, the dislocation density inside the cells was low, and there was dislocation entanglement existing on the cell wall. As shown in Figure 8b, there were sub-boundaries formed by changes in the dislocation cells.

By observation of microstructures of the matrix copper alloys and analysis of the test conditions, the dynamic recrystallization processes of the matrix copper alloys in the composites at a high temperature and high strain rate could be learned, as shown in Figure 9. Firstly, due to dynamic compression at high temperatures and strain rates, the impact pressure was very high and large plastic deformation occurred in the composites. The numbers of actuated dislocation sources thus increased and the numbers of dislocations generated by every dislocation source increased accordingly. Therefore, rather high density dislocations occurred in the matrix; forming the cellular dislocation structure constituted by tangling dislocations (A1). Then, as the stored energy was high in the center of the dislocation entanglement region, the nuclei of dynamic recrystallization first formed. Dislocation annihilation and restructuring occurred inside these dislocation entanglements, which started to transform into sub-grain boundaries (A2). With more dislocations piled and annihilated at sub-grain boundaries, the phase differences of sub-grain boundaries became larger and larger, big angular boundaries formed, and the orientations between grains became more and more random (A3). Along with the recrystallization, large, recrystallized grains of copper finally formed (A4). Under high-power TEM, we also be observed that a large number of nanometer precipitates and dislocation rings existed inside the recrystallized grains. As shown in Figure 8c, the dislocation rings formed after dislocations circumvented the mechanism and the precipitates by Orowan. Figure 8d shows the high-resolution electron microscope (HREM) morphology of the Al_3_Ni precipitate in Figure 8c,e; which is the selected area diffraction pattern (SADP) in the Figure 8d region.

## 4. Conclusions

High-density dislocations, stacking faults, twins, and recrystallized microstructures of copper alloy existed in W_f_/Cu_82_Al_10_Fe_4_Ni_4_ composites after compression with a high strain rate. High-density dislocations, stacking faults, and twins were generated due to the significant plastic deformation of copper alloy matrix under dynamic load impact, while recrystallized microstructures of copper were generated due to the dynamic recrystallization of the copper alloy matrix under dynamic compression at high temperatures.

The formation of these microstructures was closely related to the strain rate and temperature of dynamic compression. With increases in strain rate and temperature, great changes took place in the internal structure of the matrix, including tangling locations, location annihilation, cellular locations, and sub-grain boundaries. These changes led to the emergence of high-density dislocations, stacking faults, twins, and recrystallized microstructures of copper alloy.

## Figures and Tables

**Figure 1 materials-14-05563-f001:**
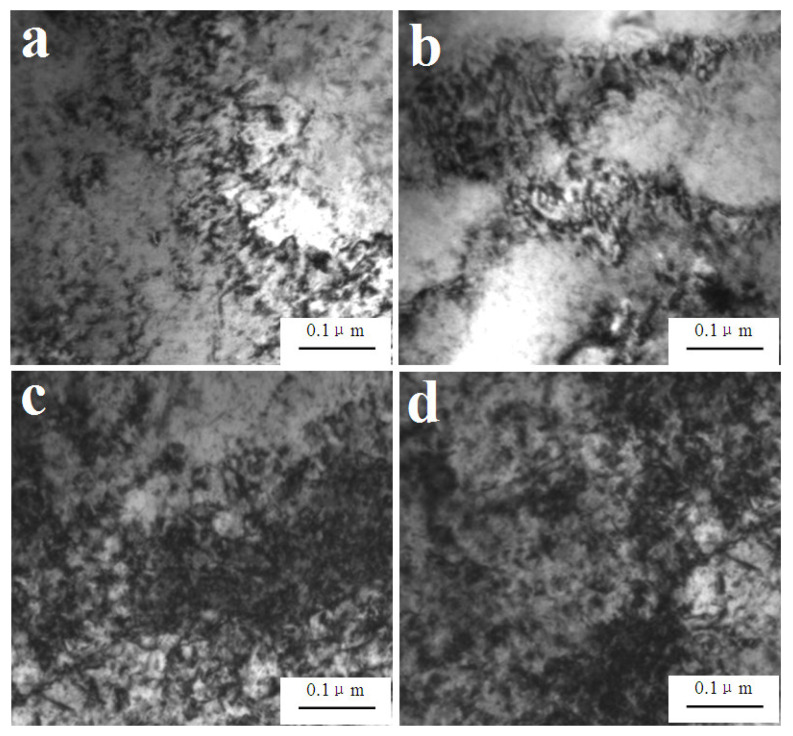
Dislocations in the matrix of W_f_/Cu_82_Al_10_Fe_4_Ni_4_ composites after dynamic compression at 20 °C: (**a**,**b**) 800 s^−1^ and (**c**,**d**) 1600 s^−1^.

**Figure 2 materials-14-05563-f002:**
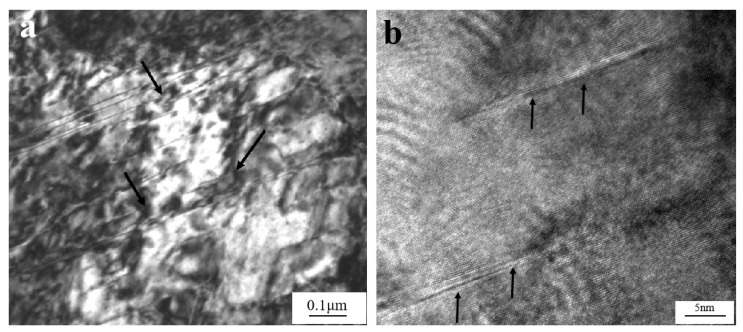
(**a**) Stacking faults in the matrix of W_f_/Cu_82_Al_10_Fe_4_Ni_4_ composites after dynamic compression at 20 °C with the strain rate of 1600 s^−1^ and (**b**) HREM morphology of stacking faults within the matrix of W_f_/Cu_82_Al_10_Fe_4_Ni_4_ composites.

**Figure 3 materials-14-05563-f003:**
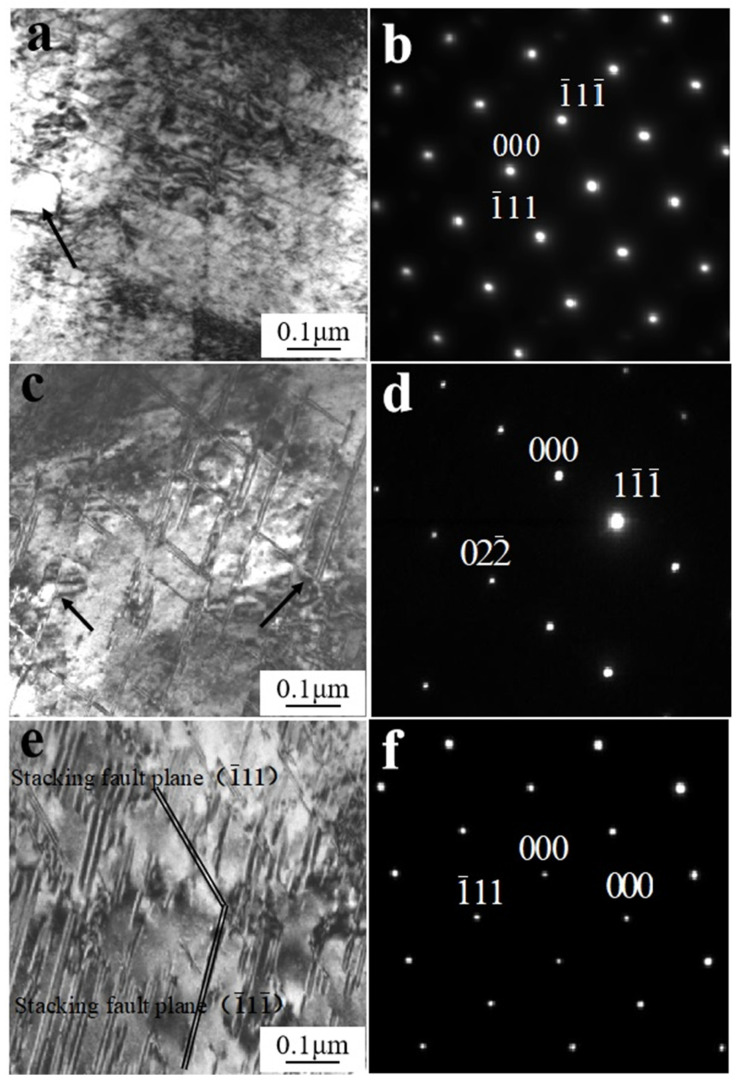
Stacking faults in the matrix of W_f_/Cu_82_Al_10_Fe_4_Ni_4_ composites after dynamic compression at high temperatures: (**a**) 200 °C, 1600 s^−1^; (**b**) SADP of (**a**); (**c**) 400 °C, 1600 s^−1^; (**d**) SADP of (**c**); (**e**) 600 °C, 1600 s^−1^; (**f**) SADP of (**e**).

**Figure 4 materials-14-05563-f004:**
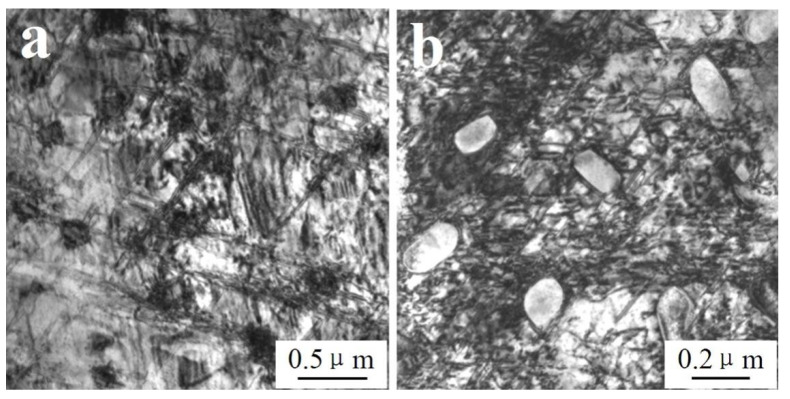
Stacking faults in the area containing lots of precipitates in matrix of W_f_/Cu_82_Al_10_Fe_4_Ni_4_ composites after dynamic compression at high temperatures: (**a**) Al_3_Ni phase and (**b**) (Fe, Ni) phase.

**Figure 5 materials-14-05563-f005:**
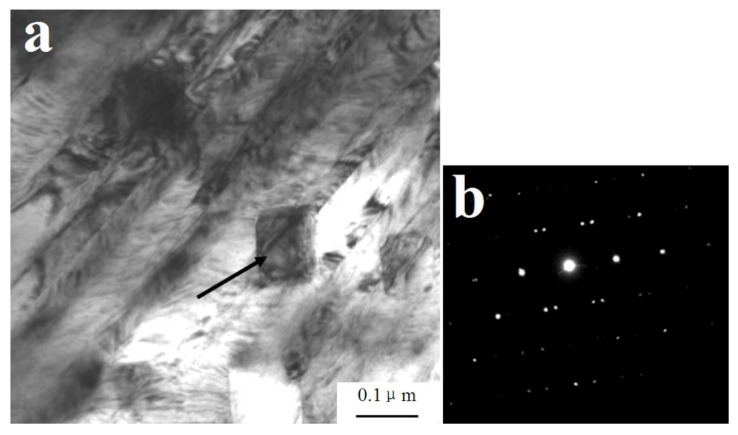
Twins in the matrix of W_f_/Cu82Al_10_Fe_4_Ni_4_ composites after dynamic compression at 400 °C with the strain rate of 1600 s^−1^: (**a**) twins and (**b**) SADP of twins.

**Figure 6 materials-14-05563-f006:**
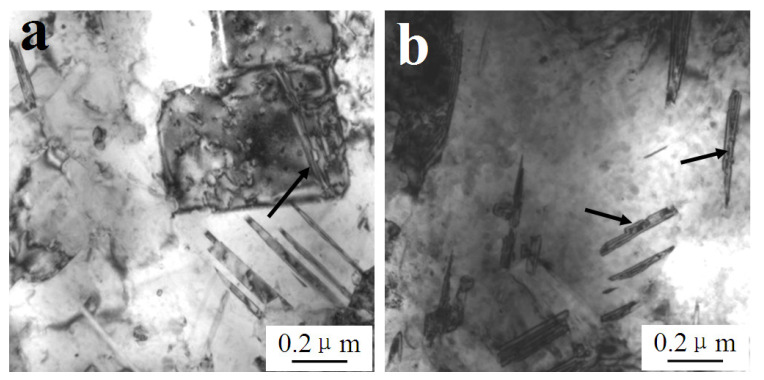
Annealing twins in the matrix of W_f_/Cu_82_Al_10_Fe_4_Ni_4_ composites: (**a**) annealing twins in recrystallized grains and (**b**) annealing twins in the matrix.

**Figure 7 materials-14-05563-f007:**
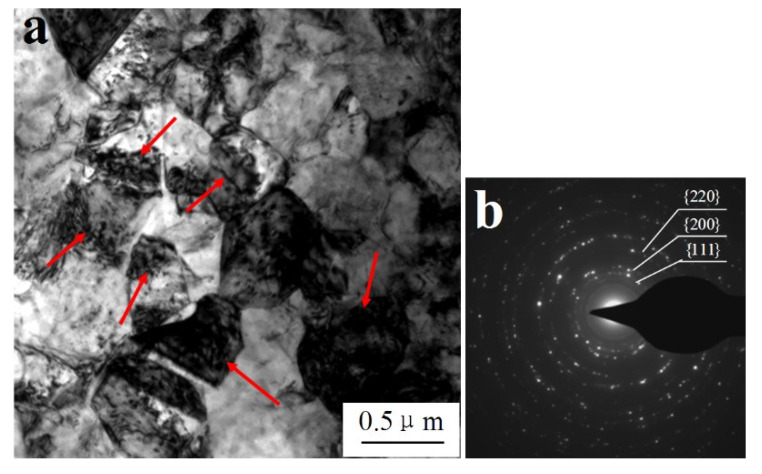
Recrystallized grains in the Cu matrix of W_f_/Cu_82_Al_10_Fe_4_Ni_4_ composites: (**a**) recrystallized grains and (**b**) SADP of recrystallized grains.

**Figure 8 materials-14-05563-f008:**
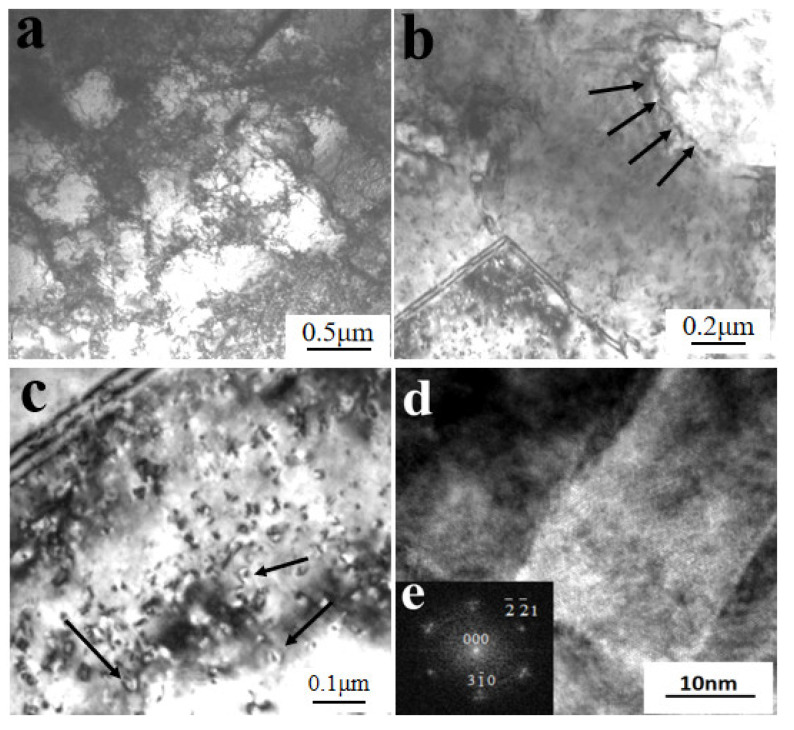
TEM microstructure of the matrix of W_f_/Cu_82_Al_10_Fe_4_Ni_4_ composites: (**a**) dislocation cells, (**b**) sub-grain-boundary, (**c**) precipitates and dislocation loops, (**d**) HREM morphology of Al_3_Ni, and (**e**) SADP of Al_3_Ni.

**Figure 9 materials-14-05563-f009:**
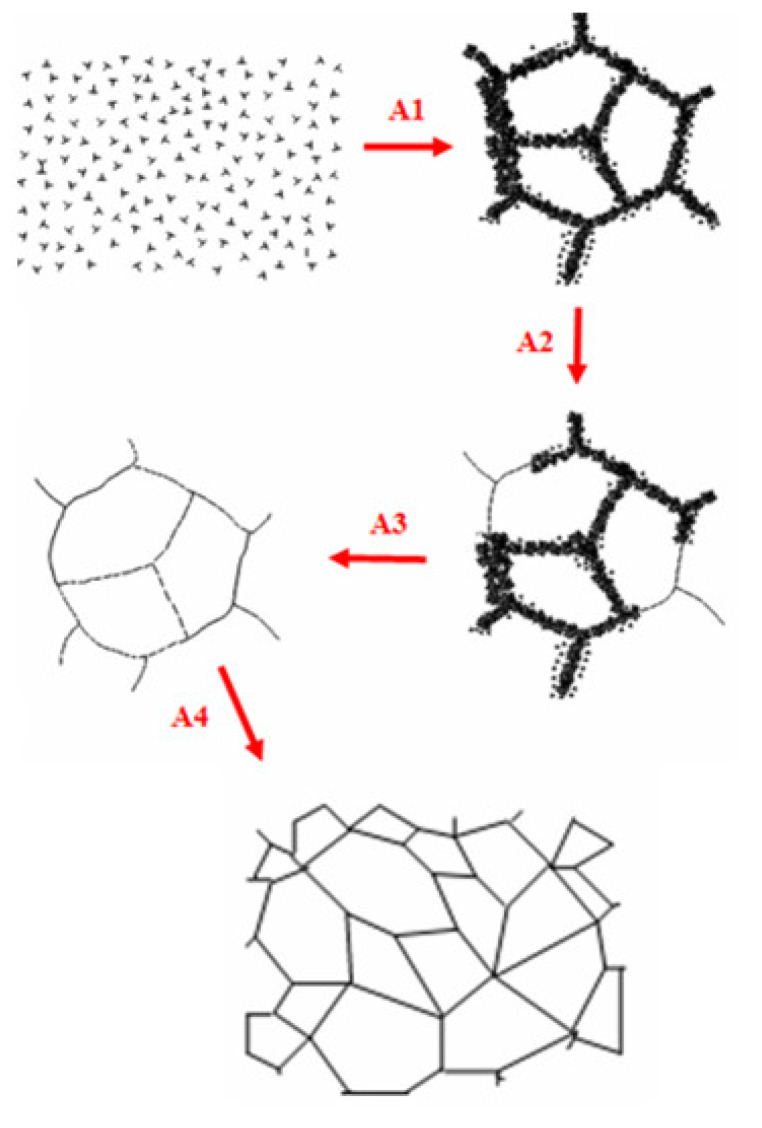
Schematic diagram of dynamic recrystallization in the matrix of W_f_/Cu_82_Al_10_Fe_4_Ni_4_ composites.

## Data Availability

Data available in a publicly accessible repository that does not issue DOIs.

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
