# Peer review of "Microstructure Evolution Mechanism of Wf/Cu82Al10Fe4Ni4 Composites under Dynamic Compression at Different Temperatures and Strain Rates"

_materials, 2021, doi:10.3390/ma14195563_

Round 1
Reviewer 1 Report
The presented manuscript deals with the microstructure evolution of W-fibers/Cu-alloy composites after dynamic loading. While the experiments are well designed and performed, the manuscript can be improved.
The introduction about the materials is well written, but there should be more information about the used methods - especially about pressure infiltration method, what are its advantages, and there should be several references to previous studies on different composites, etc. Also more details about the experiments (SHPB, TEM, diffraction) could be provided.
There are several minor errors in language and style, for example in lines 68, 100, 167, 234, or in some figures captions. Also the meaning of all abbreviations (e.g. SADP) should be written in the text. The sentences in lines 109, 117, 141 and 218 are too long and hard to read, so they should be separated or rephrased.
On the other hand, conclusion is quite short and it should be expanded to better summarize the presented work.
After resolving these issues, the manuscript could be then accepted for publication.
Author Response
Point 1: The introduction about the materials is well written, but there should be more information about the used methods - especially about pressure infiltration method, what are its advantages, and there should be several references to previous studies on different composites, etc. Also more details about the experiments (SHPB, TEM, diffraction) could be provided.
Response 1:The modification has been made according to the comments of reviewers and marked in blue and red in the paper. The specific modification position is in line 51-54 and 63-68 of the paper.
Point 2: There are several minor errors in language and style, for example in lines 68, 100, 167, 234, or in some figures captions. Also the meaning of all abbreviations (e.g. SADP) should be written in the text. The sentences in lines 109, 117, 141 and 218 are too long and hard to read, so they should be separated or rephrased.
Response 2:The modification has been made according to the comments of reviewers and marked in blue in the paper.
Point 3: On the other hand, conclusion is quite short and it should be expanded to better summarize the presented work.
Response 3:The modification has been made according to the comments of the reviewer and marked in blue in the paper. The conclusion is expanded, and the contents of the line 273-278 are added.
Please see the attachment,and the specific modifications are reflected in the uploaded attachments. Thank you.

Reviewer 2 Report
Microstructure evolution mechanism of Wf/Cu82Al10Fe4Ni4 composites under dynamic compression at different temperature and strain rate
- How can authors justify the chosen matrix composition? From the introduction it appears that authors prepared a mixture of Cu, Ni, Al, Fe in certain values and fused them together. At least the literature references reporting the characteristics and advantages of Cu82Al10Fe4Ni4 alloy are required.
- Lines 53-54: “the master alloy was set above the tungsten fibers, then Wf/Cu82Al10Fe4Ni4 composites was fabricated by means of pressure infiltration method” What was the parameters of melting?
- I recommend the authors to add the schemes of the samples’ preparation and testing
- Lines 70-71: “Secondly, large amounts of precipitates in the Cu alloy like (Fe, Ni), AlFe, Al3Ni and Cu3Al. (Fe, Ni), AlFe and Al3Ni” How did authors revealed the mentioned phases? There are neither computer simulation results nor TEM-EDS, SEM-EDS, XRD results presented in the study. Moreover, any particles in Figure 1 cannot be distinguished
- Lines 130-132: “The observed phenomenon in the 130 test was that the density of stacking faults increased with the test temperature increasing, 131 the reason for which had a lot to do with the plastic deformation of the composites.” Authors should discuss the influence of the temperature on the stacking fault energy [Phys. Rev. B 98, 224106 https://doi.org/10.1103/PhysRevB.98.224106]
- The authors reported in introduction the high industrial application of tungsten fiber composites. However, no information on the properties of studied material was given in the manuscript. I recommend the authors restructuring the introduction to emphasize the basic character of the performed research
Minor comments
Lines 194-196“As shown in Fig. 7a, after dynamic compression at 600℃, there were many large grains existing in local areas of Wf/Cu82Al10Fe4Ni4 composite sample and the grain sizes were 200-500nm.” The strain rate is not indicated
Line 64 “3.1. Microstructure of matrix with a Strain rate of 800s-1’ whereas the figure 1 shows the structures for both “a b 800s-1 . c d 1600s-1 .”
Lines 10-11, 59 “800s- 10 1 and 1600s-1” superscript should be corrected
Line 43 “therefore the dynamic dynamic compression of Wf/Cu82Al10Fe4Ni4” must be corrected
Line 50 “40% HF liquor’ solution is better
Author Response
Point 1: How can authors justify the chosen matrix composition? From the introduction it appears that authors prepared a mixture of Cu, Ni, Al, Fe in certain values and fused them together. At least the literature references reporting the characteristics and advantages of Cu82Al10Fe4Ni4 alloy are required.
Response 1: The modification has been made according to the comments of the reviewer. The reference 29 and 30 are added to explain the choice of matrix alloy
Point 2: Lines 53-54: “the master alloy was set above the tungsten fibers, then Wf/Cu82Al10Fe4Ni4 composites was fabricated by means of pressure infiltration method” What was the parameters of melting?
Response 2: The modification has been made according to the comments of reviewers and marked in red in the paper. The specific modification position is in line 66-68 of the paper.
Point 3: I recommend the authors to add the schemes of the samples’ preparation and testing
Response 3: In line 56-76 of the paper, the preparation process, testing methods and conditions of the sample are described in detail
Point 4: Lines 70-71: “Secondly, large amounts of precipitates in the Cu alloy like (Fe, Ni), AlFe, Al3Ni and Cu3Al. (Fe, Ni), AlFe and Al3Ni” How did authors revealed the mentioned phases? There are neither computer simulation results nor TEM-EDS, SEM-EDS, XRD results presented in the study. Moreover, any particles in Figure 1 cannot be distinguished
Response 4: In the previous research, the author has measured that the particles in the matrix alloy are (Fe, Ni), AlFe, Al3Ni and Cu3Al. The specific research results are described in detail in references 29 and 30. The author adds the references to the paper.
Point 5: Lines 130-132: “The observed phenomenon in the 130 test was that the density of stacking faults increased with the test temperature increasing, 131 the reason for which had a lot to do with the plastic deformation of the composites.” Authors should discuss the influence of the temperature on the stacking fault energy [Phys. Rev. B 98, 224106 https://doi.org/10.1103/PhysRevB.98.224106]
Response 5: According to the requirements of the reviewer, the author cited the required reference 35, and added a description of the effect of temperature on stacking fault density between lines 168-171.
Point 6: The authors reported in introduction the high industrial application of tungsten fiber composites. However, no information on the properties of studied material was given in the manuscript. I recommend the authors restructuring the introduction to emphasize the basic character of the performed research
Response 6: According to the requirements of the reviewer, the author revised the introduction, added lines 44-48, and marked it red.
Minor comments
Point 7: Lines 194-196“As shown in Fig. 7a, after dynamic compression at 600℃, there were many large grains existing in local areas of Wf/Cu82Al10Fe4Ni4 composite sample and the grain sizes were 200-500nm.” The strain rate is not indicated
Response 7: According to the requirements of the reviewer, the author revised it and marked the changes in red.
Point 8: Line 64 “3.1. Microstructure of matrix with a Strain rate of 800s-1’ whereas the figure 1 shows the structures for both “a b 800s-1 . c d 1600s-1 .”
Response 8: "3.1. Microstructure of matrix with a strain rate of 800s-1" mainly introduces the microstructure under 800s-1 strain rate, and the picture of 1600s-1 strain rate at this position is mainly to compare the dislocation changes under different strain rates. The purpose of putting two pictures at different strain rates together is mainly for better comparison.
Point 9: Lines 10-11, 59 “800s- 10 1 and 1600s-1” superscript should be corrected
Response 9: According to the requirements of the reviewer, the author revised it and marked the changes in red.
Point 10: Line 43 “therefore the dynamic dynamic compression of Wf/Cu82Al10Fe4Ni4” must be corrected
Response 10: According to the requirements of the reviewer, the author revised it and marked the changes in red.
Point 11: Line 50 “40% HF liquor’ solution is better
Response 11: According to the requirements of the reviewer, the author revised it and marked the changes in red.
Please see the attachment,and the specific modifications are reflected in the uploaded attachments. Thank you.
